# Exposing and Exploiting Fine-Grained Block Structures for Fast and Accurate Sparse Training

**Peng Jiang**
The University of Iowa
Iowa City, USA
peng-jiang@uiowa.edu

**Lihan Hu**
The University of Iowa
Iowa City, USA
lihan-hu@uiowa.edu

**Shihui Song**
The University of Iowa
Iowa City, USA
shihui-song@uiowa.edu

## Abstract

Sparse training is a popular technique to reduce the overhead of training large models. Although previous work has shown promising results for nonstructured sparse models, it is still unclear whether a sparse model with structural constraints can be trained from scratch to high accuracy. In this work, we study the dynamic sparse training for a class of sparse models with shuffled block structures. Compared to nonstructured models, such fine-grained structured models are more hardware-friendly and can effectively accelerate the training process. We propose an algorithm that keeps adapting the sparse model while maintaining the active parameters in shuffled blocks. We conduct experiments on a variety of networks and datasets and obtain positive results. In particular, on ImageNet, we achieve dense accuracy for ResNet50 and ResNet18 at 0.5 sparsity. On CIFAR10/100, we show that dense accuracy can be recovered at 0.6 sparsity for various models. At higher sparsity, our algorithm can still match the accuracy of nonstructured sparse training in most cases, while reducing the training time by up to 5x due to the fine-grained block structures in the models.

## 1 Introduction

As large neural networks keep advancing the state-of-the-art in many machine learning tasks, training the models has become increasingly expensive. It is reported that training GPT-3 costs an estimated $12 million in computational resources [41]. To reduce the cost, there is a growing interest in developing sparse training algorithms [30, 2, 8, 18, 26, 27]. By computing on a small subset of parameters throughout the training process, these algorithms aim to reduce the memory and computing power consumption for large-scale neural network training.

The existing sparse training methods can be categorized as either static or dynamic. While training a static sparse model from scratch is appealing, it usually requires nontrivial initialization to achieve good accuracy [9, 23, 22]. In contrast, dynamic sparse training uses simple random initialization. In every few iterations, it drops a small number of 'unimportant' weights and replaces them with some new parameters. By exploring a larger set of parameters than the sparse model itself, dynamic sparse training, in general, achieves higher accuracy than static sparse training [14].

Most of the existing dynamic sparse training methods focus on nonstructured models. That is, the active parameters are scattered in the model without any constraints. Although they can retain high accuracy with much fewer parameters, the nonstructured sparse models do not necessarily accelerate the training process because the irregular computation suffers from poor data locality and low parallelism [13, 39]. A common way to improve the efficiency of sparse models is by imposing structures on model parameters. However, current structured models have only been shown attainable by pruning and mostly beneficial to inference [13, 39, 28, 1]. Whether structured models can be obtained by sparse training and in turn accelerate the training process is still an open problem.

36th Conference on Neural Information Processing Systems (NeurIPS 2022).

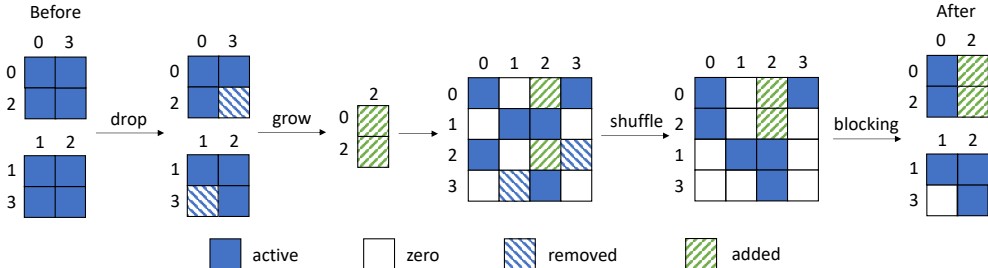

Figure 1: Model adaptation in shuffled-block dynamic sparse training: The active weights are organized in shuffled blocks before adaptation. Any weight can be dropped but new weights must be added in blocks. The new model is reorganized to ensure that all active weights are in shuffled blocks after adaptation.

Some recent work has explored *fine-grained structures* in sparse models for efficient training. For example, Hubara *et al.* [17] propose a method to find N:M transposable sparsity that can benefit both the forward and backward pass. However, their method relies on specialized hardware (sparse tensor cores). Chen *et al.* [3] test the lottery ticket hypothesis [9] for sparse models with shuffled block structures. Although they show promising results that such semi-structured models can be trained to high accuracy from scratch, how to generate a winning ticket without dense training remains unclear.

In this work, we take a step forward and investigate the existence of fine-grained block structures in sparse models with dynamic sparse training. Our idea is illustrated in Fig. 1. We first initialize a sparse model and group the parameters in shuffled blocks. Since the shuffled blocks and their transposes can be efficiently processed by hardware, our method accelerates both the forward and backward pass. In every few iterations, we adapt the model by dropping the smallest weights and adding back the same number of new parameters. Different from previous work which selects new parameters based on dense gradients [8] or dense weights [18], we select blocks of new parameters directly based on the input value and output gradient of each layer, making our algorithm purely sparse. The adapted model is shuffled and blocked again to ensure that the active weights are always organized in shuffled blocks.

We test our algorithm on a variety of networks and datasets and answer the following questions:

1. *Can a randomly initialized sparse model with fine-grained block structures be trained to high accuracy?* We train a ResNet18 and a ResNet50 of $0.5$ sparsity with parameters constrained in shuffled blocks of size $16 \times 16$ on the ImageNet dataset, and we recover the accuracy of dense models. When the sparsity increases to $0.75$, the accuracy drops slightly but still matches the accuracy of nonstructured models with dynamic sparse training. The result is further validated with various models on the CIFAR10/100 dataset.

2. *Does the model adaptation of dynamic sparse training help improve the accuracy of sparse models with fined-grained block structures?* We train sparse models with parameters in shuffled blocks with and without model adaptation. For ResNet18 and ResNet50 on ImageNet, the dynamic approach achieves more than 1% higher accuracy than static training. For different models on CIFAR10/100, dynamic training also achieves higher accuracy than static training, and the improvement is more noticeable when the sparsity is higher.

3. *What is the tradeoff between accuracy and hardware efficiency with different block sizes?* We train sparse models with different block sizes on CIFAR10/100. As expected, models with smaller block structures achieve higher accuracy. We find that the accuracy difference is small for block size 4, 8, 16 and is more noticeable for block size 32. The sparse operations on blocks of size $16 \times 16$ can be more than 3x faster than on blocks of size $4 \times 4$, but the performance improvement is small when the block size increases to 32. This suggests that block size 16 can achieve a good tradeoff between accuracy and efficiency.

To the best of our knowledge, our work is the first to apply shuffled blocking of model parameters to dynamic sparse training. We believe that our findings can inspire future research on designing hardware-efficient sparse training algorithms.

## 2   Related Work

**Neural network pruning.** The conflict between the ever-increasing size of neural networks and the limited computing capacity of hardware has inspired a lot of research on neural network pruning. Early works include [11, 10, 42, 35], in which iterative and heuristic pruning methods with limited and nonstructured sparsity were proposed. These works focus on reducing the size of neural networks but do not consider their performance carefully. The performance issue with nonstructured pruning is later identified in [40], where the authors propose to prune the entire input or output channels so that they can be easily accelerated on CPU and GPU. The downside of these structured pruning methods is that they suffer from notable accuracy loss when sparsity increases. To combine the benefits of structured and nonstructured pruning, hybrid pruning strategies have been proposed. Ma *et al.* propose pattern-based kernel pruning [29] for convolutional neural networks. The convolution kernels can only be pruned to one of several pre-defined patterns so that the pruned model contains some regular structures. Zhu *et al.* propose vector-wise pruning [45]. They divide the weight matrix into fixed-size vectors and keep the top-K elements in each vector. In addition to vector-wise pruning, the authors also adapt the tensor core architecture on GPUs to sparse tensor core in order to support the efficient execution of their vector-wise pruned models. Rumi *et al.* propose the first fine-grained block-structured pruning method for convolutional neural networks [36]. They reorder the convolution kernel matrix with a hypergraph partitioning procedure and group the nonzero weights into a number of shuffled blocks. The authors also provide a CUDA implementation of the shuffled-block-based convolution operation, and show its performance advantage over nonstructured sparse models and dense models on GPU. Recently, the shuffled blocking technique is extended for exploiting dense tensor cores on GPU to further accelerate the computation [16]. Some structured pruning methods can also reduce the computation in each iteration of the training process. However, these methods usually require much more training iterations to recover accuracy. For example, Hrank [24] prunes the convolution layers one-by-one and needs to fine-tune the model for 30 epochs after every layer is pruned. ABCPruner [25] performs end-to-end fine-tuning, but it needs to run the entire training process for multiple cycles. ResRep [7] avoids fine-tuning, but it needs to add extra "compactor" parameters for selecting the channels, resulting in extra computation in each iteration. Moreover, these structured pruning methods require a pre-trained model as input.

**Sparse training.** Pruning generates sparse models that are fast for inference; however, it does not accelerate the training process. In fact, pruning often needs to be conducted iteratively and increases the training overhead. As neural networks keep growing larger and training becomes prohibitively expensive, people start to be interested in exploiting sparsity to reduce the training cost. Mocanu *et al.* propose the first dynamic sparse training algorithm, Sparse Evolutionary Training (SET), which starts training with a randomly initialized sparse model, prunes weights with small magnitude and adds back weights at random periodically [30]. Following SET, Bellec *et al.* propose Deep Rewiring which augments traditional stochastic gradient descent with a random walk in parameter space [2]. Mostafa *et al.* propose Dynamic Sparse Reparameterization (DSR) which allows the parameter budget to shift between different layers [31]. Dettmers *et al.* propose Sparse Networks from Scratch (SNFS) which uses the momentum of each parameter as the criterion for growing weights [5]. Evci *et al.* point out that all these algorithms can be expressed with a drop-and-grow framework and propose RigL which uses the gradient magnitude as the grow criterion [8]. An issue with RigL is that it needs to compute the full gradient every few iterations, which may not be affordable for large models. Top-kast addresses this issue by maintaining all the weights but computing gradients on a small subset of weights in each iteration. Although these sparse training algorithms show promising results, they still have an accuracy gap with pruning methods. Recently, researchers show that this gap can be closed by using a small batchsize and training for more iterations [26] or by interleaving sparse training with dense training phases [34]. Besides dynamic sparse training, there has been a line of research on static sparse training. Lee *et al.* propose Single-Shot Network Pruning (SNIP) which finds an initial mask with one-shot pruning and keeps the sparse model fixed during training [23]. In a follow-up paper, they refine their initialization method from a signal propagation perspective [22]. Frankle *et al.* propose the Lottery Ticket Hypothesis (LTH) which states that a sparse model can be trained to match the accuracy of the dense model if a good initialization is provided [9]. While appealing, the effectiveness of LTH in reducing the training cost is still unclear as obtaining the "winning ticket" is usually nontrivial [44, 9].

Most of the existing sparse training algorithms (either static or dynamic) focus on nonstructured sparse models. Although they reduce the number of floating-point operations, these models cannot

run efficiently on hardware due to their irregular memory access patterns and limited parallelism. This puts the practicality of sparse training algorithms into doubt. Inspired by the hybrid pruning methods, researchers have worked on extracting fine-grained structures in sparse models to accelerate training. Hubara *et al.* propose a method to find N:M transposable sparsity [17]. Their method accelerates both the forward and backward pass in training and can be used in either a static or dynamic fashion. However, the N:M sparsity relies on specialized hardware (sparse tensor cores on GPU) to deliver good performance and thus is not generally applicable to other hardware or even older GPUs. Chen *et al.* [3] adopt the shuffled blocking technique from [36] and show that a sparse model with shuffled block structures can be trained to dense accuracy at a nontrivial sparsity ratio. However, they need to pre-train a dense model to obtain an initialization for the sparse training. Dietrich *et al.* [6] propose a dynamic sparse training algorithm with block sparsity. However, as we show in §5 and experiments, block sparsity without row/column shuffling has a very low mask diversity and achieves low model accuracy.

## 3 Preliminaries

**Notation.** Throughout the paper, we use $W \in \mathbb{R}^{R \times C}$ to denote the parameters in a layer. Here, $R$ is the output channel size; $C$ is the input channel size (if it is a linear layer), or the input channel size times the convolution kernel size (if it is a convolutional layer). We use $In \in \mathbb{R}^{C \times B}$ to denote the input and $Out \in \mathbb{R}^{R \times B}$ to denote the output, where $B$ is the batch size. The gradient w.r.t. $W$ is $dW$ and has the same dimension as $W$. The element at row $i$ and column $j$ of $dW$ is denoted as $dW_{ij}$. The gradient w.r.t the output is $dOut$, and the gradient w.r.t. the input is $dIn$. We use $|| \cdot ||$ to denote the $\ell_1$ norm of a matrix and use $| \cdot |$ to denote the absolute value of each element in a matrix or vector.

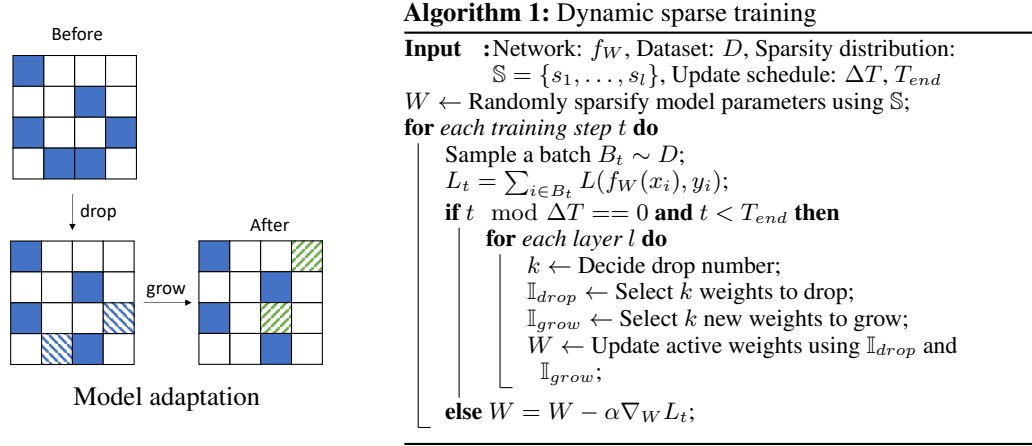

**Algorithm 1:** Dynamic sparse training

**Input** : Network: $f_W$, Dataset: $D$, Sparsity distribution:
$\mathbb{S} = \{s_1, \ldots, s_l\}$, Update schedule: $\Delta T$, $T_{end}$
$W \leftarrow$ Randomly sparsify model parameters using $\mathbb{S}$;
**for** *each training step $t$* **do**
  Sample a batch $B_t \sim D$;
  $L_t = \sum_{i \in B_t} L(f_W(x_i), y_i)$;
  **if** $t \mod \Delta T == 0$ **and** $t < T_{end}$ **then**
    **for** *each layer $l$* **do**
      $k \leftarrow$ Decide drop number;
      $\mathbb{I}_{drop} \leftarrow$ Select $k$ weights to drop;
      $\mathbb{I}_{grow} \leftarrow$ Select $k$ new weights to grow;
      $W \leftarrow$ Update active weights using $\mathbb{I}_{drop}$ and $\mathbb{I}_{grow}$;
  **else** $W = W - \alpha \nabla_W L_t$;

Figure 3: Dynamic sparse training with drop-and-grow-based model adaptation.

**Dynamic sparse training.** Fig. 3 illustrates the idea of dynamic sparse training. The algorithm starts with a randomly initialized sparse model. It adapts the model in every $\Delta T$ iterations until iteration $T_{end}$. The model adaptation involves two operations: *drop* and *grow*. The drop operation discards some weights in the model, and the grow operation adds some new weights back. Different sparse training algorithms use different criteria for the two operations. For example, RigL [8] drops the weights with the smallest magnitudes and grows the same amount of weights with the largest gradients. Top-kast [18] maintains a superset of active weights and drops and grows active weights according to their magnitudes. With the periodic model adaptation, dynamic sparse training algorithms can explore more parameters and achieve higher accuracy than static sparse training [27, 26].

**Computations in sparse neural networks.** Many operations in a dense neural network can be expressed as general matrix multiplications (GEMM). In a sparse model, the computations are sparse matrix multiplications. For example, the training of a linear layer involves three matrix multiplications: 1) in the forward pass, the input is multiplied with the sparse weight matrix to get the output (i.e., $Out = W \cdot In$); 2) in the backward pass, the gradient of loss w.r.t. the input is computed by multiplying the transpose of the sparse weight matrix with the gradient w.r.t. the

**Algorithm 2:** Forward pass: $Out = W \cdot In$ for a shuffled block of parameters

**Input** : $W[S_1][S_2]$, $rowidx[S_1]$, $colidx[S_2]$, $In[C][B]$
**Output:** $Out[R][B]$
$In'[S_2][B] = In[colidx][B]$;
$Out'[S_1][B] = 0$;
**for** $i = 0$ **to** $S_1 - 1$ **do**
    **for** $j = 0$ **to** $S_2 - 1$ **do**
        **for** $k = 0$ **to** $B - 1$ **do**
            $Out'[i][k]$ +=
            $W[i][j] * In'[j][k]$;

$Out[rowidx][B] = Out'$;

**Algorithm 3:** Backward pass: $dW = dOut \cdot In^T$ for a shuffled block of parameters

**input** : $rowidx[S_1]$, $colidx[S_2]$, $dOut[R][B]$, $In[C][B]$
**output:** $dW[S_1][S_2]$
$In'[S_2][B] = In[colidx][B]$;
$dOut'[S_1][B] = dOut[rowidx][B]$;
$dW'[S_1][S_2] = 0$;
**for** $i = 0$ **to** $S_1 - 1$ **do**
    **for** $j = 0$ **to** $S_2 - 1$ **do**
        **for** $k = 0$ **to** $B - 1$ **do**
            $dW'[i][j]$ += $dOut'[i][k] * In'[j][k]$;

$dW[rowidx][colidx] = dW'$;

output (i.e., $dIn = W^T \cdot dOut$); 3) the gradient of loss w.r.t. the weight is computed by multiplying the gradient w.r.t. the output with the transpose of input (i.e., $dW = dOut \cdot In^T$). The first two operations are sparse-matrix dense-matrix multiplication (SpMM). The last operation is a sampled dense-dense matrix multiplication (SDDMM) because only the gradients for the active weights need to be computed.

**Sparse matrix multiplication with shuffled blocks.** SpMM and SDDMM were first studied for efficient execution on GPU in the scientific computing domain [15, 19]. They show that the performance bottleneck of the two operations is the irregular access to the input dense matrix, and they propose *adaptive tiling and reordering* techniques to improve the memory access efficiency. While they target large matrices with high sparsity, the reordering and tiling ideas have inspired *shuffled blocking* techniques to accelerate SpMM and SDDMM for smaller matrices with relatively lower sparsity in sparse neural networks [36, 16, 3]. The idea is illustrated in Fig. 4. The nonzeros in the sparse matrix are organized into small blocks. Algorithm 2 shows the implementation of the forward pass for a shuffled block of parameters. Suppose the row indices of the block are $rowidx$ and column indices are $colidx$. The program first loads $In[colidx][B]$ into cache $In'$ and allocates a cache $Out'$ for $Out[rowidx][B]$.

Since the values of $In[colidx][B]$ and $Out[rowidx][B]$ are repeatedly accessed in the computation and accessing cache is much faster than main memory, the memory access overhead is reduced. Similarly, Algorithm 3 shows the computation of the gradients for a block of parameters. The program first loads $In[colidx][B]$ and $dOut[rowidx][B]$ into cache and reuses the data in cache throughout the computation. The larger the block size, the more data

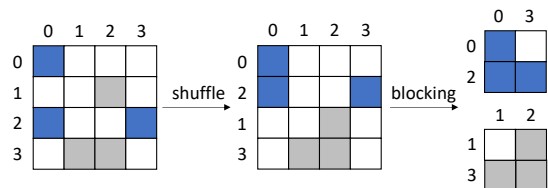

Figure 4: Shuffled blocking a sparse matrix.

reuse the program has. The computation can be further accelerated by exploiting the matrix-multiply units in emerging hardware (e.g., tensor cores on Nvidia GPU) [16]. This is why structured sparse models can deliver better performance than nonstructured models that have even fewer parameters.

# 4 Proposed Algorithm

As explained in the previous section, for sparse models to run efficiently, it is important to have all the active weights in blocks. Previous work ensures this through pruning [36, 16] or initialization [3]. We now show that sparse models with such fine-grained block structures can also be obtained by dynamic sparse training. Fig. 1 illustrates the model adaptation steps in our algorithm. This section details each of the steps.

**Initialization.** The algorithm starts with a randomly initialized nonstructured model. We adopt the ERK sparsity distribution [30] to determine the sparsity for each layer. Then, a shuffled blocking procedure is invoked to reshape the sparse model into shuffled blocks.

**Shuffled blocking.** Suppose $N$ is the number of active weights in a layer and the block size is set to $S_1 \times S_2$. The shuffled blocking procedure aims to put as many active weights into $\lceil N/(S_1 S_2) \rceil$ blocks. As shown in Fig. 4, it first reorders the rows of the sparse matrix so that similar rows are close to each other. The similarity between two rows is defined as the Jaccard similarity of the nonzero columns in the two rows (i.e., the number of common columns divided by the total number of distinct columns). Intuitively, two rows with more common columns have a larger similarity and should be put close to each other. With the definition of similarity between rows, the shuffling of rows can be achieved by a simple clustering procedure. We use the $AgglomerativeClustering$ from sklearn [33] for this task. Since row reordering is performed repeatedly during the training process, we want to keep its overhead as small as possible. We observe that most rows have small similarities with other rows, and reordering them does not improve much of the block density. Therefore, we only cluster the rows that have more than $q$ nonzeros in common columns with other rows, where $q$ (0.2 in our experiments) is a hyperparameter that controls the clustering overhead. After the rows are reordered, we divide them into groups of size $S_1$. In Fig. 4, $S_1$ is set to 2, and thus the four rows are divided into two groups: $\{0,2\}$ and $\{1,3\}$. Within each row group, we select $S_2$ columns to form a block of size $S_1 \times S_2$. To ensure large weights are preserved, we always select the column with the largest $\ell_1$ norm in each row group and return the block with the largest $\ell_1$ norm among all row groups. This procedure continues until we find enough blocks.

**Block-aware drop criterion.** The sparse model is adapted in every $\Delta T$ iterations. For each layer, we compute the number of weights to be dropped in iteration $t$ with a cosine decaying function:

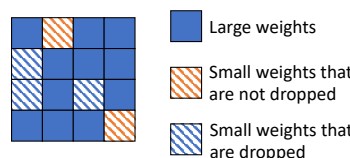

Large weights

Small weights that are not dropped

Small weights that are dropped

$$k = \frac{\alpha N}{2} \left( 1 + cos \left( \frac{t\pi}{T_{end}} \right) \right) \quad (1)$$

where $\alpha$ is the initial drop ratio and $N$ is the number of active weights in the layer. Following previous dynamic sparse training algorithms [8, 26, 27, 18], we select the $k$ weights with the smallest magnitudes (i.e. $ArgTopK(-|W|, k)$) for

Figure 5: Avoid dropping isolated small weights.

removal. Before removing them, we check if these weights make up at least half of the elements in either a row or a column of a block. More specifically, in each shuffled block $\mathcal{B}$, we have the weights to be dropped as $\mathcal{B} \cap ArgTopK(-|W|, k)$. For a weight $w \in \mathcal{B} \cap ArgTopK(-|W|, k)$, the elements in the same row of $\mathcal{B}$ is $\mathcal{B}[rowidx(w), :]$, and the elements in the same column of $\mathcal{B}$ is $\mathcal{B}[:, colidx(w)]$. We remove $w$ if and only if $\mathcal{B}[rowidx(w), :] \cap ArgTopK(-|W|, k)$ has at least $S_1/2$ elements or $\mathcal{B}[:, colidx(w)] \cap ArgTopK(-|W|, k)$ has at least $S_2/2$ elements. If a weight in $ArgTopK(-|W|, k)$ does not have enough other weights to be dropped in the same row/column of a block, the weight will not be dropped. This is because even if we drop it, the weight is very likely to be added back after the blocking procedure. However, because the number of weights to be added in the grow phase is determined by the number of dropped weights, this ineffective dropping of some small weights will result in more new weights being added in the grow phase and may cause more important weights to be discarded by the following shuffled blocking procedure. Our block-aware drop criterion reduces the negative impact of such ineffective drops. Intuitively, when the small weights are concentrated in a few rows or columns, we perform more model adaptation. When the small weights are scattered in many rows and columns, we perform less aggressive adaptation. Fig. 5 shows an example of the block-aware drop criterion. Suppose there are five small weights (marked with diagonal stripes) in a block. Instead of dropping all the five weights, we only drop three of them that are in a row or column with at least two small weights (marked in blue). The two weights with only themselves to be dropped in their rows and columns (marked in orange) will remain active. In our experiments, we find that this block-aware drop criterion is critical for achieving good accuracy for shuffled-block dynamic sparse training (See Appendix A.2).

**Block-wise grow criterion.** The main novelty of our method lies in how we grow the weights. Different from previous work that computes the gradients for all weights and adds the weights with the largest gradients[8, 26, 27], we use $dOut$ and $In$ to estimate the importance of weights. Specifically, since $dW = dOut \cdot In^T$, $dW_{ij} = \langle dOut_i, In_j \rangle$, we have $||dW_{ij}|| \leq ||dOut_i|| \cdot ||In_j||$ where $dOut_i$ represents row $i$ of $dOut$ and $In_j$ represents row $j$ of the input. This suggests that the weights connecting the input channels of larger magnitudes to the output channels of larger gradients are of greater importance. Our grow criterion is designed based on this intuition. Suppose $X$ represents a group of $S_x$ rows in $dOut$ and $Y$ represents a group of $S_y$ rows in $In$. It is easy to see that $XY^T$ is a shuffled block of size $S_x \times S_y$ in $dW$. Our goal is to find the blocks of weights with

Table 1: Mask diversity for different sparsity constraints.

| Matrix Size ($R \times C$) | $8 \times 8$ | | | $16 \times 16$ | | |
|---|---|---|---|---|---|---|
| Sparsity | 1:2 | 2:4 | 4:8 | 1:2 | 2:4 | 4:8 |
| Nonstructured | 1.8e+18 | | | 5.8e+75 | | |
| N:M | 4.3e+9 | 2.8e+12 | 5.8e+14 | 3.4e+38 | 6.3e+49 | 1.1e+59 |
| N:M Transposable | 6.6e+4 | 4.3e+8 | 1.8e+13 | 1.8e+19 | 3.4e+34 | 9.5e+52 |
| Nonshuffled Block ($S_1 = S_2 = R/4$) | 12870 | | | | | |
| Nonshuffled Block ($S_1 = S_2 = R/2$) | 6 | | | | | |
| Shuffled Block ($S_1 = S_2 = R/4$) | 1.2e+18 | | | 3.7e+48 | | |
| Shuffled Block ($S_1 = S_2 = R/2$) | 3.1e+11 | | | 1.3e+29 | | |

the largest $||XY^T||$. Without computing the actual gradients, we use $||X|| \cdot ||Y^T||$ as an estimate of gradient magnitude. We first sort rows of $dOut$ and $In$ according to their norms in descending order and divide sorted $dOut$ into groups of $S_x$ rows and sorted $In$ into groups of $S_y$. Then, we compute $||X|| \cdot ||Y^T||$ for the shuffled blocks and select the blocks with largest $||X|| \cdot ||Y^T||$. For a $dOut$ of size $x \times n$ and an input $In$ of size $y \times n$, the FLOPS for computing the norms of rows of $dOut$ and $In$ is $xn + yn$, and the FLOPS for computing the sparse gradients is no more than $kn$ where $k$ is the number of dropped weights in Formula (1). So the total FLOPS with our proposed grow criterion is $xn + yn + kn$. In comparison, the FLOPS for computing dense gradient is $xyn$. Since $k$ is small, our block-wise grow criterion has a much smaller time complexity than dense gradient computation.

Fig. 6 shows an example of the selection procedure with $S_x = 2$ and $S_y = 1$. Suppose the sorted rows of input are 2, 1, 0, 3, and the sorted rows of $dOut$ are 0, 2, 3, 1. The selection starts from the block in the top left corner ($b_0$) as it has the largest $||X|| \cdot ||Y^T||$. With $b_0$ selected, the two blocks ($b_1$ and $b_4$) adjacent to $b_0$ are added to the frontier, and the larger one between the two will be selected in the next step. The selected block may overlap with the existing active blocks. Only the non-overlapping parameters are considered new parameters. We keep selecting new blocks and updating the frontier until new parameters exceed the drop number. In the example of Fig. 1, a block of new parameters ([0,2] and [2,2]) are added. Finally, we reorganize the new model into shuffled blocks. To prevent the new parameters from being discarded by the blocking procedure, we set their value to the maximum value of the active weights. After the

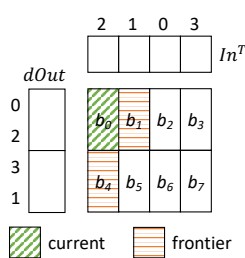

Figure 6: Select new parameters in blocks.

new blocks are formed, we reset the value of the new parameters to zero so that they will not affect the output of the network.

## 5 Comparison with Transposable N:M Sparsity

Our work is closely related to the line of research that explores fine-grained N:M sparsity in neural networks [43, 17, 38]. While most work focuses on accelerating inference, a recent paper shows that transposable N:M sparsity can be found in sparse models to accelerate both the forward and backward pass in training [17]. This section compares transposable N:M sparsity with shuffled block sparsity.

In [17], the authors propose a measure called *mask diversity* to quantify how much a specific structure constrains the model. Specifically, mask diversity is the total number of distinct sparse masks that satisfy certain sparsity and dimension constraints. They show a positive correlation between the mask diversity and accuracy. We follow their arguments and compute the mask diversity of shuffled block sparsity. For a matrix of size $R \times C$, suppose $R$ and $C$ are divisible by the block size $S_1$ and $S_2$, and the sparsity is $s$. Since the rows are reordered and divided into disjoint groups, the number of ways to choose row groups is $R!/(S_1!)^{R/S_1}$. Once the row groups are fixed, we need to select a total of $RC(1-s)/S_2$ columns from all row groups to form the desired number of blocks. This leads to the following formula for the mask diversity of shuffled block sparsity:

$$MD_{sh\_block} = \frac{R!}{(S_1!)^{R/S_1}} \cdot \binom{RC}{RC(1-s)/S_2}. \tag{2}$$

In comparison, the mask diversity of nonshuffled block sparsity is simply choosing $RC(1-s)/(S_1 S_2)$ blocks from the total $RC/(S_1 S_2)$ blocks. Table 1 lists the mask diversity of shuffled block sparsity

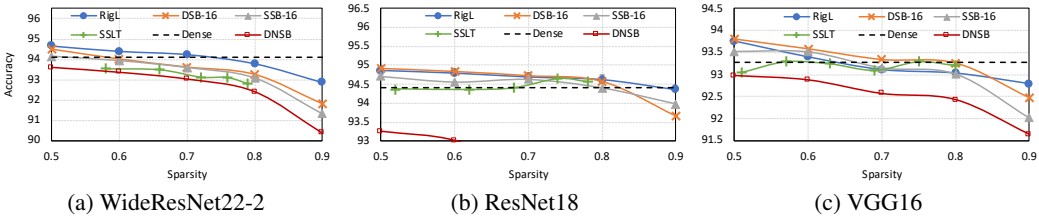

Figure 7: Test accuracy (%) over sparsity on CIFAR10.

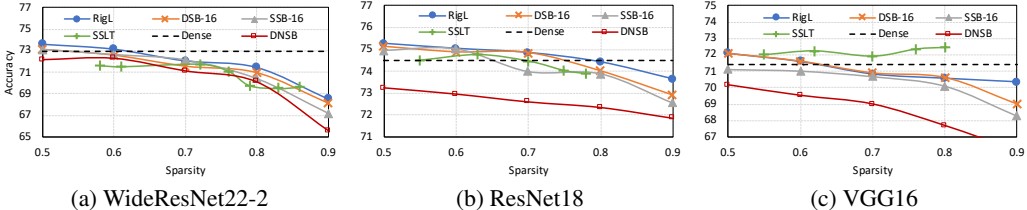

Figure 8: Test accuracy (%) over sparsity on CIFAR100.

along with other sparsity constraints for an $8 \times 8$ matrix and a $16 \times 16$ matrix. The formula for other sparsity constraints can be found in [17]. We can see that nonstructured sparsity has the highest diversity, which explains its superior accuracy to structured sparsity. When the block size is small, shuffled block sparsity has a mask diversity close to nonstructured sparsity. For example, for the $8 \times 8$ matrix, when $S_1 = S_2 = R/4 = 2$, the mask diversity is of the same order of magnitude as nonstructured sparsity. When the block size increases, the mask diversity decreases. This illustrates a tradeoff between hardware efficiency and model accuracy for shuffled block sparsity. Compared to N:M sparsity which only benefits the forward pass, shuffled block sparsity benefits both the forward and backward pass and yet has a higher mask diversity when $S_1 = S_2 \leq 4$ and $M = 2$. Compared to N:M transposable sparsity, shuffled block sparsity has a higher mask diversity when $S_1 = S_2 \leq 4$ and $M \leq 4$.

Moreover, N:M sparsity relies on sparse-matrix-multiply units in emerging hardware (sparse tensor cores on Nvidia Ampere GPU) to deliver good performance, whereas shuffled blocking accelerates the computation on traditional computing units and can exploit dense-matrix-multiply units which are more broadly supported by hardware (e.g. dense tensor cores on GPU, AMX on Intel CPU, MMA on IBM Power10 processor). As the current Nvidia GPU only supports sparse tensor cores with $N = 2, M = 4$ and dense tensor cores with $S_1 = S_2 = 4$, shuffled blocking is expected to be a more flexible and more efficient way to accelerate sparse training. We can also see that shuffled block sparsity has a much higher mask diversity than nonshuffled block sparsity. This is because the shuffling of rows and columns significantly increases the number of ways to form nonzero blocks.

## 6  Experiments

We test our algorithm with ResNet18 and ResNet50 [12] on the ImageNet dataset [4], and WideResNet22-2, ResNet18 and VGG16 [37] on CIFAR10 [20] and CIFAR100 dataset [21]. For

Table 2: Top-1 accuracy on ImageNet. DSB represents our Dynamic Shuffled Block training, and SSB represents Static Shuffled Block training with random initialization. The number before DSB and SSB indicates sparsity, and the number after DSB and SSB is the block size. SSLT represents Structured Sparse Lottery Ticket from [3]. The number before SSLT indicates sparsity. '-' indicates missing data in literature.

| Model | Dense | 0.5 DSB-16 | 0.5 SSB-16 | 0.51 SSLT | 0.7 SSLT | 0.75 DSB-16 | 0.75 SSB-16 | 0.8 RigL |
|-------|-------|------------|------------|-----------|----------|-------------|-------------|----------|
| ResNet50 | 76.13 | **76.33** | 75.83 | 75.65 | 71.5 | 74.04 | 72.76 | 75.1 |
| ResNet18 | 69.76 | **69.84** | 68.52 | - | | 65.72 | 64.49 | - |

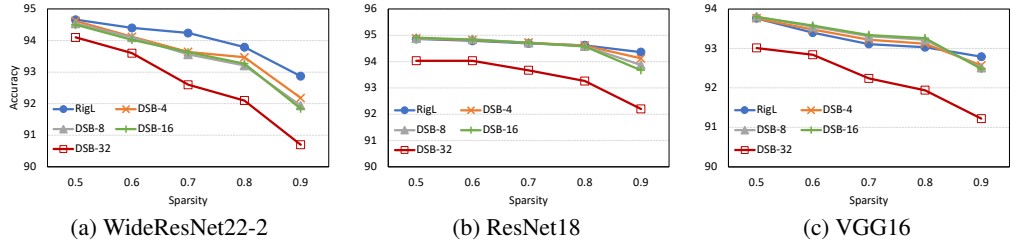

Figure 9: Test accuracy (%) over sparsity on CIFAR-10 with DSB using different block sizes. RigL corresponds to DSB using block size 1.

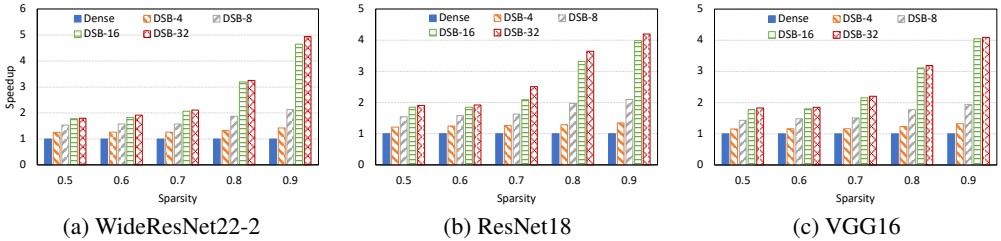

Figure 10: Average speedup of sparse convolution over dense convolution operations on CIFAR10.

ImageNet, the models are trained for 100 epochs with batch size 256. The learning rate schedule starts with a linear warm up reaching its maximum value of 0.1 at epoch 5 which is then dropped by a factor of 10 at epochs 30, 70, 90. For CIFAR10/100, the models are trained for 250 epochs with batch size 128. The learning rate is initialized to 0.1 and is decreased by a factor of 5 every 30,000 iterations. The standard SGD with a momentum of 0.9 is used as the optimizer. For dynamic sparse training, the initial drop ratio $\alpha$ is set to 0.3, $\Delta T$ is set to 1600 for ImageNet and 100 for CIFAR10/100, and $T_{end}$ is set to 400000 for ImageNet and 75000 for CIFAR10/100.

**Accuracy compared with other training methods.** Table 2 lists the top-1 accuracy of ResNet18 and ResNet50 trained with different methods on ImageNet. Our dynamic shuffled block training (DSB) recovers the accuracy of the dense models at 0.5 sparsity. In comparison, the structured sparse lottery ticket (SSLT) [3] has a slight accuracy drop for ResNet50 at 0.51 sparsity while requiring a dense pre-training for initializing the sparse mask. The accuracy of DSB at sparsity 0.75 is also higher than that of SSLT at sparsity 0.7. Fig. 7 shows the test accuracy of WideResNet22-2, ResNet18 and VGG16 trained with different methods on CIFAR10. All the experiments are run five times. The accuracies plotted in the figures are the average values. The difference in accuracy between different runs is smaller than 0.05%. We can see that our DSB recovers dense accuracy at 0.5 and 0.6 sparsity for all three models. In contrast, SSLT has a noticeable accuracy drop for WideResNet22 and ResNet18 (although they can achieve dense accuracy at lower sparsities [3]). Compared with nonstructured dynamic sparse training (RigL [8]), our DSB has slightly lower accuracy at 0.5 and 0.6 sparsity, and the accuracy gap widens as the sparsity gets higher. Fig. 8 shows the test accuracy on CIFAR100. Again, our DSB recovers dense accuracy at 0.5 and 0.6 sparsity. For WideResNet22-2 and ResNet18, our DSB achieves slightly higher accuracy than SSLT at small sparsity ($\leq 0.7$) and almost the same accuracy at higher sparsity ($> 0.7$). For VGG16, SSLT achieves surprisingly higher accuracy than both dense training and our DSB, but they require a dense pre-training for initialization. For all the test cases, nonshuffled block sparsity (DNSB) achieves apparently lower accuracies than other methods, which is consistent with the mask diversity shown in Table 1.

**Importance of model adaptation.** The results in Table 2 and Fig. 7 and 8 also show that the model adaptation is critical to the accuracy of sparse training. For ResNet50 and ResNet18 on ImageNet, without model adaptation, static shuffled block training (SSB) cannot recover dense accuracy at 0.5 sparsity. Its accuracy at 0.75 sparsity is more than 1% lower than DSB. For WideResNet22-2, ResNet18, and VGG16 on CIFAR10/100, SSB also has lower accuracy than DSB, and the accuracy gap is larger at higher sparsity.

**Tradeoff between accuracy and efficiency with different block sizes.** To show the effects of block size on accuracy, we run DSB with different block sizes. The results are shown in Fig. 9. DSB-4,8,16 achieve almost the same accuracy when sparsity $\leq$ 0.8. DSB-4 shows a slight advantage over DSB-8,16 at 0.9 sparsity. When the block size increases to 32, we observe an apparent accuracy drop, and the drop is more significant at higher sparsity. To show the effects of block size on performance, we test the implementation of shuffled block convolution from [36] on an Nvidia RTX3090 GPU. Fig. 10 shows the average speedups of shuffled block convolution with different block sizes over dense convolution. We can see that when the block size is small, the sparse convolution has almost no performance advantage over dense convolution, even at a high sparsity. As the block size increases, the shuffled block convolution achieves better performance. Notably, when the block size increases from 8 to 16, the sparse computation achieves the largest marginal speedups. When the block size increases from 16 to 32, the performance improvement is small. This is because the convolution operation with $16 \times 16$ blocks already saturates the cache on the GPU platform. The results suggest that block size 16 achieves a good tradeoff between the accuracy and efficiency of DSB.

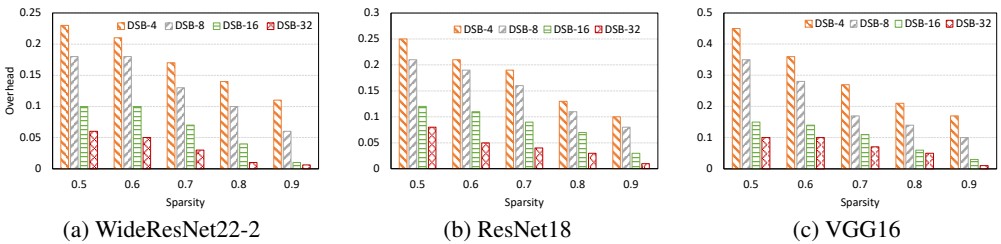

(a) WideResNet22-2        (b) ResNet18        (c) VGG16

Figure 11: The ratio of shuffled blocking time to dense training time for different models on CIFAR10.

**Overhead of shuffled blocking.** Although shuffled blocking is an expensive operation, it is performed only once in every $\Delta T$ iterations. Fig. 11 shows the overhead of shuffled blocking. The execution time of shuffled blocking accounts for 1% to 45% of total training time. We can see that, as the sparsity increases, the overhead of shuffled blocking decreases. As the block size increases, the number of blocks decreases, and the overhead also decreases. When block size is set to 16, the shuffled blocking procedure takes about 10% of the total training time. According to the speedups in Fig. 10, the overhead can be justified by the performance improvements brought by sparse computation.

## 7 Conclusion

We propose a dynamic sparse training algorithm that extracts and exploits fine-grained block structures in sparse models. Our algorithm is designed based on the drop-and-grow model adaptation framework and features a block-aware drop criterion and a block-wise grow criterion. We show that a randomly initialized sparse model with shuffled block structures can be trained to high accuracy, and the training process can be effectively accelerated.

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
