

(a) WideResNet22-2      (b) ResNet18      (c) VGG16

Figure 12: Training time per epoch for different models on CIFAR10.

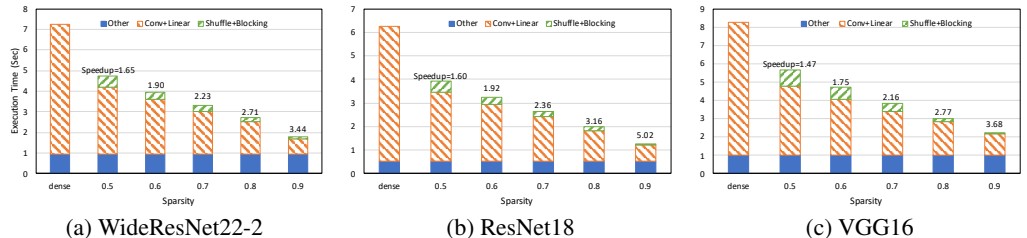

(a) WideResNet22-2      (b) ResNet18      (c) VGG16

Figure 13: Training time per epoch for different models on CIFAR100.

# A   Appendix

## A.1   End-to-end speedups

To show the end-to-end speedups with shuffled-block sparsity, we obtain the layer-wise execution time of dense training on an Nvidia RTX 3090 GPU with PyTorch Profiler [32]. The sparse training time is estimated by replacing the execution time of dense convolution and linear operations with the execution time of shuffled-block convolution and linear operations. Fig. 12 shows the average training time per epoch for WideResNet22-2, ResNet18, and VGG16 on CIFAR10. The batch size is set to 128, and the shuffled block size is set to 16. We can see that convolution and linear operations account for more than 80% execution time of the dense models. Shuffled-block sparse training effectively reduces the execution time of these layers at different sparsities, achieving overall 1.46x to 5.02x speedups. Fig. 13 shows similar speedups for the three models on CIFAR100 dataset.

## A.2   Benefits of block-aware drop criterion

Figure 14 shows the accuracy of shuffled-block dynamic sparse training with and without our block-aware drop criterion for WideResNet22-2, ResNet18, and VGG16 on CIFAR10 dataset. As we explain in §4, the original drop criterion leads to ineffective drops of small weights and causes the more important weights to be discarded by the shuffled blocking procedure. The experiments show that the original drop criterion has apparently lower accuracies than our block-aware drop criterion. Fig. 15 shows a similar pattern for the three models on CIFAR100 dataset. The results validate the advantage of our block-aware drop criterion for shuffled-block dynamic sparse training.

## A.3   Memory consumption analysis

Table 3 lists the memory consumption of our dynamic shuffled-block training (DSB) and two other dynamic sparse training methods (RigL [8] and Top-KAST [18]). We only compare the memory consumption of model weights and gradients since the intermediate activations are of the same size with different training methods. Suppose the number of parameters in the dense model is $N$ and the density of the sparse model is $d$ $(0 < d \le 1)$. In each training iteration, RigL [8] stores $Nd$ weights and computes their gradients. For model adaptation, RigL needs to compute the gradients for all the $N$ parameters and select the parameters with the largest gradients for growing. Our DSB has the same memory consumption as RigL in each training iteration, but we need not compute the full gradients for model adaptation thanks to our block-wise grow criterion. Top-KAST [18] aims to

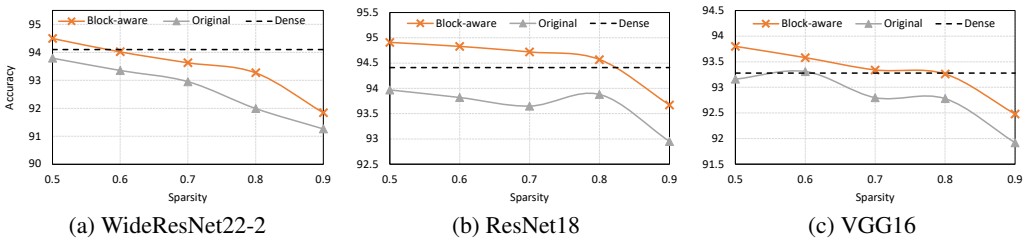

(a) WideResNet22-2  (b) ResNet18  (c) VGG16

Figure 14: Test accuracy (%) of shuffled-block dynamic sparse training with and without our block-aware drop criterion on CIFAR10.

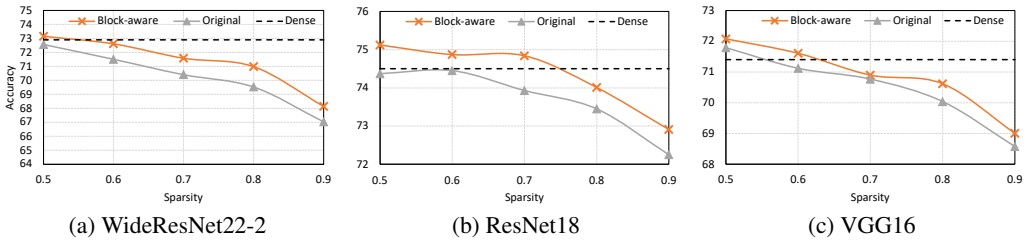

(a) WideResNet22-2  (b) ResNet18  (c) VGG16

Figure 15: Test accuracy (%) of shuffled-block dynamic sparse training with and without our block-aware drop criterion on CIFAR100.

avoid the computation of full gradients for model adaption. It computes the gradients for a superset of the active weights (with $ND + M$ parameters) in each iteration. After updating the $ND + M$ parameters, it selects $ND$ parameters with the largest magnitudes from all parameters as the active weights for the next iteration. Thus, it needs to store all the $N$ parameters. Compared to RigL and Top-KAST which needs to store either dense gradients or dense weights, our DSB stores sparse gradients and weights throughout the training process.

Table 3: Memory consumption of model weights and gradients with different sparse training methods. $N$ is the number of parameters in the dense model, $d$ represents the density of the sparse model, and $M$ represents the extra number of gradients that need to be computed in Top-KAST. "Train" refers to the memory consumption in each training iteration, and "Model adaptation" refers to the memory used for adapting the sparse model.

|  | Train | | Model adaptation | |
|---|---|---|---|---|
|  | weights | gradients | weights | gradients |
| DSB | $Nd$ | $Nd$ | $Nd$ | $Nd$ |
| RigL | $Nd$ | $Nd$ | $Nd$ | $N$ |
| Top-KAST | $N$ | $Nd + M$ | - | - |