# OpenReview forum: "Exposing and Exploiting Fine-Grained Block Structures for Fast and Accurate Sparse Training"
_NeurIPS.cc/2022/Conference — NeurIPS 2022 Accept_

### Official Review · Reviewer_pyCy · 2022-07-06

**Rating:** 6
**Confidence:** 4
**Soundness:** 3 good
**Presentation:** 3 good
**Contribution:** 3 good

**Summary:**

This paper explores sparse training for fine-grained block sparse models. Detailedly, the authors leverage the classic prune-and-regrow workflow in a block-aware manner with a new grow criterion that does not rely on dense gradient calculation of weights. Experiments on ImageNet and CIFAR-10/100 show that the proposed block-wise sparse training can achieve satisfying performance and noticeable speedup at moderate sparse rates.

**Questions:**

1. The authors claim that DSB accelerates the computation on traditional computing units that are broadly supported by hardware. However, I notice that their provided codebase is a stimulated version, i.e., weights are multiplied with binary masks to represent pruned model. It would be nice for the authors to provide a running demo or code that can be deployed on general hardware, which could also make this paper more convincing and easy-to-implement.

**Ethics Review Area:**

["I don’t know"]

**Limitations:**

The authors do not discuss the limitations of their work as well as the negative societal impact.

**Strengths And Weaknesses:**

Strengths:
1. The reviewer appreciates the motivation and contribution of this work. The exploration of hardware-friendly sparse training is remarkable and could benefit the community a lot.
2. The proposed block-wise growing criterion is well-designed and can effectively reduce the overhead of sparse training, despite that the ablation studies are missing. I recommend the authors compare traditional dense gradient criterion with the proposed block-wise grow criterion to further demonstrate their view.
3. This paper is well-written and easy to follow.

Weakness:
1. The authors claim that the proposed block-wise sparsity can match the performance of unstructured sparsity in most cases. However, DSB seems to lag far behind the current state-of-the-art unstructured methods. For example, STR[1] achieves 76.12% top-1 accuracy at 81% sparsity, while DSB only achieves 74% top-1 accuracy at 75% sparsity.
2. The core advantage of the proposed DSB falls in acceleration and computation reduction on general hardware. However, the authors only compare DSB with unstructured methods. In contrast, structured sparsity methods[2,3,4] can also eliminate the training and inference costs. Thus, it is necessary to add comparisons with structured methods to show whether the DSB can obtain a better trade-off between speedup and performance maintenance.

Minor:
1. What is SDB In Tab.2?  I recommend the authors add a relevant caption.

[1] Soft Threshold Weight Reparameterization for Learnable Sparsity. In ICML,2020.
[2] Hrank: Filter pruning using high-rank feature map. In CVPR, 2020.
[3] Resrep: Lossless cnn pruning via decoupling remembering and forgetting. In ICCV, 2021
[4] Channel Pruning via Automatic Structure Search. In IJCAI, 2020.

---

> ### Author Response · Authors · 2022-08-02
> **Response to Reviewer pyCy**
>
> > Q1: The authors claim that the proposed block-wise sparsity can match the performance of unstructured sparsity in most cases. However, DSB seems to lag far behind the current state-of-the-art unstructured methods. For example, STR[1] achieves 76.12% top-1 accuracy at 81% sparsity, while DSB only achieves 74% top-1 accuracy at 75% sparsity.
>
> A1: Note that STR[1] is a nonstructured **pruning** method. It focuses on obtaining a sparse model for fast inference, but it requires dense computation in the training phase (for updating the reparameterized weights). In contrast, our work focuses on speeding up the training procedure by training a sparse model from scratch. The observation we made in this work is that **shuffled-block dynamic sparse training can match the performance of nonstructured dynamic sparse training**. It is expected our accuracy will be lower than nonstructured pruning, as most nonstructured dynamic sparse training methods still cannot match the accuracy of nonstructured pruning at high sparsity.
>
>
>
> > Q2: It is necessary to add comparisons with structured methods to show whether the DSB can obtain a better trade-off between speedup and performance maintenance.
>
> A2: Thank you for the suggestion. We have added a qualitative comparison with the listed structured pruning methods in the Related Work section. The main difference between our work and the previous structured pruning methods is that **we start the training with a randomly initialized model and do not require extra training iterations.** In contrast, all the three structured pruning methods require a pre-trained model as input. Hrank [1] performs layer-wise pruning and needs to fine-tune the model for 30 epochs after every layer is pruned. ABCPruner [2] performs end-to-end fine-tuning, but it still needs to run the entire training process for multiple cycles. ResRep [3] avoids fine-tuning, but it adds extra "compactor" parameters for selecting the channels, introducing extra computation in each iteration.
>
>
>
> > Q3:What is SDB In Tab.2?
>
> A3:It should be DSB. We have corrected it.
>
>
>
> > Q4: The authors claim that DSB accelerates the computation on traditional computing units that are broadly supported by hardware. However, I notice that their provided codebase is a stimulated version, i.e., weights are multiplied with binary masks to represent pruned model...
>
> A4: We used the CUDA implementation of shuffled-block convolution from a previous work, which is publicly available online: https://github.com/VITA-Group/Structure-LTH/tree/main/profile/regroup_conv. Please also see **Reviewer RD2v Q1**.
>
>
>
>
>
> [1] Kusupati et al. Soft Threshold Weight Reparameterization for Learnable Sparsity. In ICML,2020.
>
> [2] Lin et al. Hrank: Filter pruning using high-rank feature map. In CVPR, 2020.
>
> [3] Lin et al. Channel Pruning via Automatic Structure Search. In IJCAI, 2020.
>
> [4] Ding et al. Resrep: Lossless cnn pruning via decoupling remembering and forgetting. In ICCV, 2021.

---

> > ### Comment · Reviewer_pyCy · 2022-08-08
> > **Response**
> >
> > Thanks for your reply, I will keep my score.

---

### Official Review · Reviewer_vhN2 · 2022-07-09

**Rating:** 4
**Confidence:** 4
**Soundness:** 2 fair
**Presentation:** 2 fair
**Contribution:** 3 good

**Summary:**

This paper proposes a hardware-friendly sparse training method with shuffled block structures, which can be accelerated by the common hardware (GPU). The acceleration can be achieved by quickly obtaining the index of the grouped weights during the forward and backward pass. The research topic is very important for the sparse training community. In spite of the promising results of dynamic sparse training (DST), a major problem of DST is that the promising efficiency can not be fully transferred to the common hardware. Therefore, any good contribution to the topic is important. However, several major concerns need to be clarified as written below.

**Questions:**

Please refer to the above weakness.

**Strengths And Weaknesses:**

## Strengths

(1) This paper proposes a structured sparse training method by grouping the unstructured elements together during training. It is surprising to see that the grouped DST can still match the performance of the dense model while achieving up to 5x acceleration.

(2) The criterion used in the paper does not require to calculate the dense gradients.

(3) Related work is relatively extensive.

##  Weakness

(1) The authors claim that they are the first to apply group sparsity to dynamic sparse training. However, I think they have missed one prior work (https://arxiv.org/pdf/2108.06277.pdf) which has explored this direction, even though the method is distinct. I suggest adjusting their claim.

(2) The description of the methodology is relatively dry and hard to follow. More effort should be put to improve it.

(3) My major concern is that the authors did not mention which part of the results are produced with masks and which part of the results are produced with the proposed method? It has been claimed in the abstract that the algorithm can match the accuracy while reducing training time by 5x, but no actual wall-clock training time has been reported in the paper. More specifically, (1) does the proposed method support end-to-end dynamic sparse training? As far as I understand, the dense matrix implementation in Pytorch or Tensorflow is so well-optimized, that even we can achieve acceleration by quickly selecting the index of grouped weights, it is difficult to be quicker than the mask-version DST in terms of the real wall-clock time. (2) Was the averaged speed up reported in figure 9 be evaluated with one feed-forward pass or evaluated with the end-to-end training? This question is important to verify the real contribution of the proposed method.

(4) I am curious about the real speedup of the proposed grow criterion? It would be nice to provide how many flops can be saved by using this instead of calculating the dense gradient? In reality, how fast it is compared with dense gradient on GPU?

---

> ### Author Response · Authors · 2022-08-02
> **Response to Reviewer vhN2**
>
> > Q1: missed one prior work...
>
> A1: Thank you for the reference. We have added a discussion of this work in the Related Work section. The main difference between our work and this prior work is that we use shuffled block structure while they use nonshuffled block structure. As we explained in **Reviewer RD2v Q2 and Q3**, this shuffling step is critical for achieving high accuracy. We also proposed a block-aware drop criterion to improve the accuracy of shuffled-block dynamic sparse training, and a block-wise grow criterion to reduce the gradient computation.
>
>
>
> > Q2: The description of the methodology is relatively dry and hard to follow.
>
> A2: We have added more details and an example (Fig.5) in Section 4 to help explain our ideas.
>
>
>
> > Q3: My major concern is that the authors did not mention which part of the results are produced with masks and which part of the results are produced with the proposed method?
>
> A3: All of the accuracy results (Fig.7,8,9) are obtained by simulation with masks. All of the execution times (Fig.10,11) are obtained by real CUDA implementation of shuffled-block convolution. We have added the end-to-end training time in the supplementary. Please also see **Reviewer RD2v Q1**.
>
>
>
> > Q4: I am curious about the real speedup of the proposed grow criterion? It would be nice to provide how many flops can be saved by using this instead of calculating the dense gradient?
>
> A4: For a $dOut$ of size $x\times n$ and an input In of size $y\times n$, the FLOPS for computing the norms of rows of dOut and In is $xn + yn$, and the FLOPS for computing the sparse gradients is no more than $kn$ where $k$ is the number of dropped weights in Formula (1). So the total FLOPS with our proposed grow criterion is $xn+yn+kn$. In comparison, the FLOPS for computing dense gradient is $xyn$. Since $k$ is small, our time complexity is much smaller than dense gradient computation. We have added the comparison in Section 4.

---

> > ### Author Response · Authors · 2022-08-08
> > **Sincerely expecting further discussions with Reviewer vhN2**
> >
> > Dear reviewer vhN2,
> > Many thanks for your review of our paper. Considering the discussion period deadline, we look forward to receiving your feedback on our response. If you have any other questions about the paper, please also let us know.

---

> > > ### Comment · Reviewer_vhN2 · 2022-08-08
> > > **Followup**
> > >
> > > I thank the authors for providing their responses.
> > >
> > > After reading the response, I believe my concern about the end-to-end training speedup has not been addressed. The way the paper used to report the real training time is still simulation (replacing the execution time of each dense convolution layer with the time of shuffled block convolution and the overhead of periodic shuffled blocking). If the proposed method indeed supports the end-to-end real training acceleration, it would be natural to report both the accuracy and the training time with the CUDA implementation? Could the authors explain more about this?

---

> > > > ### Author Response · Authors · 2022-08-09
> > > > **About the end-to-end speedup evaluation**
> > > >
> > > > We understand your concern about the end-to-end evaluation. We want to point out that our method for evaluating the end-to-end speedup with layer-wise CUDA execution time is commonly used in the literature. According to the source code of [1] and [2] and our conversation with the authors, they evaluate the end-to-end execution time by replacing the layer-wise execution time of dense convolution with shuffled-block convolution. Another work on accelerating shuffled-block sparse model with tensor cores [3] also uses layer-wise CUDA execution time to estimate overall speedups. A closely related work on sparse training with N:M sparsity [4], which we compared with in the paper, does not even have a layer-wise CUDA implementation, and they estimate speedups based on the percentage of computation that can benefit from the N:M sparsity.
> > > >
> > > > The reason for such an evaluation of end-to-end speedup is that integrating the sparse operators into deep learning systems such as PyTorch requires much more engineering effort. Before the system researchers or industry put effort into implementing end-to-end sparse training systems, we should be able to investigate sparse training algorithms with simulated evaluation. The overall speedup is guaranteed as long as the percentage of computation that can benefit from sparse operations and the speedup realized by each sparse operation are validated.
> > > >
> > > > [1] Rumi et al. Accelerating Sparse CNN Inference on GPUs with Performance-Aware Weight Pruning. In PACT 2020.
> > > > [2] Chen et al. Coarsening the Granularity: Towards Structurally Sparse Lottery Tickets. In ICML 2022.
> > > > [3] Huang et al. Shfl-BW: Accelerating Deep Neural Network Inference with Tensor-Core Aware Weight Pruning. In DAC 2022.
> > > > [4] Hubara et al. Accelerated Sparse Neural Training: A Provable and Efficient Method to Find n: m Transposable Masks. In NeurIPS 2021.

---

### Official Review · Reviewer_eD4a · 2022-07-11

**Rating:** 6
**Confidence:** 3
**Soundness:** 3 good
**Presentation:** 3 good
**Contribution:** 3 good

**Summary:**

This paper studies dynamic sparse training (DST) with fine-grained structured pruning.
Starting with a randomly initialized nonstructured model, the shuffled blocking procedure groups active weights into several blocks. During the training interval, it drops some weights with the smallest magnitudes, and grows some weights with the proposed block-wise grow criterion. With experiments on CIFAR and ImageNet, this paper demonstrates that the proposed framework is better than structured sparse lottery ticket (SSLT) [3] and RigL [6].


**Questions:**

This paper is an overall OK submission. Nevertheless, the lack of acceleration comparison refrains me to give a better score. I would be happy to improve my rating as long as the authors address the above weaknesses in the rebuttal.


**Limitations:**

No, this paper does not discuss the limitation and potential negative societal impact.

**Strengths And Weaknesses:**

Strengths

- This paper firstly studies the DST with fine-grained structured models. Compared with previous works, the proposed method no longer requires specialized hardware (sparse tensor cores on GPU) [16] or pre-trained dense models [3].
- The proposed growing algorithm is novel and practical as it avoids the calculation of dense gradients and dense weights, which makes the training purely sparse.
- This paper clearly discusses the difference between the proposed method and the Transposable N: M Sparsity [16].
- The paper is in general well written and easy to follow.

Weaknesses

- More details about the hardware acceleration should be provided.
Since this paper focuses on sparse training with fine-grained structured pruning, the total training time saving and memory saving should be reported. Meanwhile, this paper should compare the speedup with some structured pruning methods in Figure 9.

- In Table 2, Figure 6, and Figure 7, the error bar should be presented.

---

> ### Author Response · Authors · 2022-08-02
> **Response to Reviewer eD4a**
>
> > Q1: More details about the hardware acceleration should be provided. Since this paper focuses on sparse training with fine-grained structured pruning, the total training time saving and memory saving should be reported.
>
> A1: We have added the end-to-end training speedups in the supplementary (Please see our response to **Reviewer RD2v Q1**). We have also added an analysis of memory consumption with different sparse training methods.
>
>
>
> > Q2: This paper should compare the speedup with some structured pruning methods...
>
> A2: It is true that some structured pruning methods can reduce the computation in each iteration of the training process. However, these methods usually require much more training iterations to recover accuracy. Take the structured pruning methods listed by **Reviewer pyCy** as an example, Hrank [1] prunes the convolution layers one by one and needs to fine-tune the model for 30 epochs after every layer is pruned. ABCPruner [2] allows end-to-end fine-tuning, but it still needs to run the entire training process for multiple cycles. ResRep [3] avoid fine-tuning, but it needs to add extra "compactor" parameters for selecting the channels, introducing extra computation in each iteration. Moreover, all these pruning methods require a pre-trained model as input. Our sparse training method, in contrast, aims to reduce the computation of training for a **randomly initialized model without increasing the number of training iterations**. We have added a discussion of these structured pruning methods in the Related Work section.
>
>
>
> > Q3: In Table 2, Figure 6, and Figure 7, the error bar should be presented.
>
> A3: We repeated the experiments in Fig.6,7 for five times. The difference in accuracy between different runs is smaller than 0.05%. We tried to plot the error bar in the figures, but the error is so small that it cannot show clearly on the figures. We have clarified this in the paper.
>
>
>
> [1] Lin et al. Hrank: Filter pruning using high-rank feature map. In CVPR, 2020.
>
> [2] Lin et al. Channel Pruning via Automatic Structure Search. In IJCAI, 2020.
>
> [3] Ding et al. Resrep: Lossless cnn pruning via decoupling remembering and forgetting. In ICCV, 2021.

---

> > ### Comment · Reviewer_eD4a · 2022-08-08
> > **Thanks for your reply.**
> >
> > The response addresses my most concerns. Thus, I keep my positive rating.

---

### Official Review · Reviewer_RD2v · 2022-07-12

**Rating:** 5
**Confidence:** 3
**Soundness:** 2 fair
**Presentation:** 2 fair
**Contribution:** 2 fair

**Summary:**

The paper proposes a sparse training method based on fine-grained block sparsity. The main insight is that shuffled block sparsity (where entries in the weight matrices are permuted to form blocks) can be hardware efficient and yield speed up. The method proposed consists of block-pruning and block- growing at regular intervals (i.e., dynamic block sparsity). Both block-pruning and block-growing can be done efficiently without computing gradient for the whole dense matrix. Experiments on ResNets & VGG on CIFAR and ImageNet show that at 0.50 sparsity, the proposed method can match the accuracy of the dense models.

**Questions:**

1. What is the overhead of block dropping? The paper claims it improves accuracy but there's no experiments to show this.


**Limitations:**

Limitations are discussed.

**Strengths And Weaknesses:**

Strengths:
1. The block-wise grow criterion is clever and well-motivated. It avoids having to compute the gradient for the entire dense matrix. Instead, it estimates the gradient magnitude by multiply the magnitude of the output gradient and the input. Maybe micro-experiment to show how close this heuristic is compared to the expensive gradient of the dense matrix would make it even more convincing.
2. The accuracy vs block sizes ablation is informative. It reveals that block size of 16x16 can yield speedup while maintaining accuracy of the dense model.

Weaknesses:
1. Lack of validation on wallclock-time speedup. Does the benchmark use channel first or channel last? fp32 or fp16? These details significantly change the speed of the dense model.
What is the overall model speedup, either in inference or training?
The code seems to just use mask to multiply with a dense weight matrix so I'm not sure how one could get wall-clock time speedup.
2. Lack of comparison with existing block-sparse methods (e.g., Block Pruning For Faster Transformers. Lagunas et al. EMNLP 2021). Does the shuffled block sparsity perform better / faster than just block sparsity?
3. The concept of shuffled block should to be explained more clearly. Is the entire 16x16 block either zero or nonzero? Or are there both zeros and nonzeros in the same block?
How important is the shuffle step? How does one decide which rows / columns to shuffle?

---

> ### Author Response · Authors · 2022-08-02
> **Response to Reviewer RD2v**
>
> > Q1: Lack of validation on wallclock-time speedup...What is the overall model speedup...
>
> A1: We have uploaded a supplementary document with the end-to-end speedups. As we explained in the experiments, we used the CUDA implementation of shuffled-block convolution from [1] for execution time evaluation. The same implementation was used by a recent work on shuffled-block lottery tickets [2], and their source code is publicly available at https://github.com/VITA-Group/Structure-LTH/tree/main/profile/regroup_conv. It is a channel first, FP32 implementation. The layer-wise speedups in Fig.10 (Fig.9 in the original version) were collected with this code.
>
> To obtain the end-to-end training time, we profiled dense model training using PyTorch profiler, replaced the execution time of each dense convolution layer with the time of shuffled block convolution, and added the overhead of periodic shuffled blocking. This estimated execution time should be close to the real end-to-end execution time of sparse training with shuffled-block convolution.
>
> With that said, we want to clarify that implementing an end-to-end sparse training system is not the focus of this work, as the hardware efficiency of shuffled-block sparsity has been proven in previous works for both inference [1] and training [2]. Instead, the main contribution of this work is that we show dynamic sparse training with shuffled-block structures can achieve higher accuracy than static sparse training with shuffled-block lottery tickets (SSLT [2]) and achieves comparable accuracy to nonstructured dynamic sparse training (RigL [3]), providing a new direction for investigating hardware-efficient dynamic sparse training algorithms.
>
>
>
> > Q2: Lack of comparison with existing block-sparse methods...Does the shuffled block sparsity perform better / faster than just block sparsity?
>
> A2: Thank you for the suggestion. We have added the accuracy of nonshuffled-block dynamic sparse training in Fig.7 and 8. We also added the mask diversity of nonshuffled block sparsity in Table1. Both the experimental results and the mask diversity show a clear advantage of the shuffled block sparsity.
>
>
>
> > Q3: The concept of shuffled block should be explained more clearly. Is the entire 16x16 block either zero or nonzero? Or are there both zeros and nonzeros in the same block? How important is the shuffle step?
>
> A3: The entire block is either zero or nonzero. Note that the elements in a shuffled block are scattered in non-contiguous rows and columns of a sparse matrix (as shown in Fig.3). This shuffling step significantly improves the mask diversity of the sparse model compared with nonshuffled blocks. We have added a comparison of mask diversity with nonshuffled block sparsity in Section 5.
>
>
>
> > Q4: How does one decide which rows / columns to shuffle?
>
> A4: As we explained in Section4 **Shuffled blocking**, the shuffling of rows is performed by a clustering procedure. To keep its overhead small, we only shuffle the rows that have similarities with other rows larger than a threshold. We have added more details about the shuffling procedure in Section 4.
>
>
>
> > Q5: What is the overhead of block dropping? The paper claims it improves accuracy but there's no experiments to show this.
>
> A5: The claim in the original version might be confusing. We proposed a block-aware drop criterion in Section 4. It achieves better accuracy for shuffled-block dynamic sparse training than the original drop criterion used by nonstructured dynamic sparse training algorithms because it avoids ineffective dropping of small weights. We have rephrased the claim to "our block-aware drop criterion reduces the negative impact of ineffective drops with the original drop criterion." We have also added an example (Fig. 5) to help explain the idea, and added an experimental comparison with the original drop criterion in the supplementary.
>
>
>
> [1] Rumi et al. Accelerating Sparse CNN Inference on GPUs with Performance-Aware Weight Pruning. In PACT 2020.
>
> [2] Chen et al. Coarsening the Granularity: Towards Structurally Sparse Lottery Tickets. In ICML 2022.
>
> [3] Evci et al. Rigging the Lottery: Making All Tickets Winners. In ICML 2020.

---

> > ### Author Response · Authors · 2022-08-08
> > **Sincerely expecting further discussions with Reviewer RD2v**
> >
> > Dear reviewer RD2v,
> > Thank you for your careful review of our original paper. As the discussion period will end soon, we hope to see if our response addresses your concerns and if you have further questions.

---

> > > ### Comment · Reviewer_RD2v · 2022-08-08
> > > **Updates**
> > >
> > > Thank you for the thorough response. The end-to-end speedup explanation is very helpful. I've adjusted my rating.

---

### Author Response · Authors · 2022-08-02
**Response to all reviewers**

We thank the reviewers for many suggestions on the evaluation and presentation of results. We have significantly revised the paper according to the feedback. The new additions are highlighted in the pdf, but a quick summary of the changes is below:

1. We added a supplementary document with an evaluation of end-to-end speedups, benefits of block-aware drop criterion, and a comparison of memory consumption with other sparse training methods.
2. More details and an example are added in Section 4 to help explain our main ideas.
3. An analysis of time complexity for our block-wise grow criterion is added in Section 4.
4. A comparison with nonshuffled block sparsity in terms of mask diversity and model accuracy is added in Section 5 and Fig.7,8.
5. Discussions of structured pruning methods and a previous work on dynamic sparse training with nonshuffled block sparsity are added in the Related Work section.

We hope these can address the reviewers' concerns, and we look forward to discussing further with the reviewers.

---

### Public Comment · ~Vladimír_Boža1 · 2024-10-20
**Code uses dense gradients (contrary to claims in the paper)**

Paper claims:
```
Different from previous work which selects new parameters based on dense gradients [8] or dense weights [18], we choose blocks of new parameters directly based on the input value and output gradient of each layer, making our algorithm purely sparse.
```

But when we look at the code (convolution.py), we find that the growing code uses `grad_without_mask = torch.mul(self.weight.grad, torch.logical_not(self.weight_mask)).detach()`, which is a dense gradient!
Moreover, code actually records layer inputs and gradients of layer outputs (using forward and backward hooks), but never uses them!

So, the proposed method for growing weights is actually never experimentally verified, and all of the results are done using dense gradients.

Can authors explain this discrepancy?

---

> ### Public Comment · Authors · 2024-10-21
> **No full gradient is computed.**
>
> Thank you for your interest in our work. We believe the code you are looking for is in linear.py line 83 to 104. It uses the norms of rows/columns of the input and dout to select the row/columns of gradients to computes, so no full gradient is computed.
> We later found that dout and input do not need to be explicitly stored in forward and backward hooks because the row/column numbers can be directly computed in the hooks. However, the github repo has not been maintained since. We will upload the code with the better implementation.

---

### Meta-Review · Area_Chair_EDrP · 2022-08-29

**Recommendation:** Accept
**Confidence:** Certain

**Metareview:**

This paper studies dynamic sparse training (DST) with fine-grained structured pruning. A topic of practical relevance, the authors proposed hardware-friendly shuffled blocking and simultaneously prune-and-grow. While this paper seems to be a mild extension from Chen et.al. 2022, the contributions are clear. One main concern raised is whether evaluating the end-to-end speedup with layer-wise CUDA execution time is a correct practice. I agree with the authors, that is common in the ML literature and more real evaluation requires system-level engineering (often beyond the scope of an algorithm paper).

Three out of four reviewers acknowledged they were convinced by the authors' rebuttal. One reviewer mainly questioned one missing reference (while acknowledging “distinct”) + several algorithm step clarifications, to which the authors delivered point-by-point responses. Therefore, given the overall sentiment among reviewers, I’m recommending accept.

**Award:**

No

---

### Decision · Program_Chairs · 2022-09-14

Accept